# Effect of HY7602 Fermented Deer Antler on Physical Fatigue and Antioxidant Activity in Mice

**DOI:** 10.3390/ijms25063318

**Published:** 2024-03-14

**Authors:** Hyejin Jeon, Kippeum Lee, Yong-Tae Kim, Joo-Yun Kim, Jae-Jung Shim, Jae-Hwan Lee

**Affiliations:** R&BD Center, HY Co., Ltd., 22, Giheungdanji-ro 24beon-gil, Giheung-gu, Yongin-si 17086, Republic of Korea; 10003012@hy.co.kr (H.J.); joy4917@hanmail.net (K.L.); christianyt@hy.co.kr (Y.-T.K.); jjshim@hy.co.kr (J.-J.S.); jaehwan@hy.co.kr (J.-H.L.)

**Keywords:** anti-fatigue, antioxidant, fermentation, deer antler, lactic acid bacteria, probiotics, *Lactobacillus curvatus*, HY7602

## Abstract

*Lactobacillus curvatus* HY7602 fermented antler (FA) ameliorates sarcopenia and improves exercise performance by increasing muscle mass, muscle fiber regeneration, and mitochondrial biogenesis; however, its anti-fatigue and antioxidant effects have not been studied. Therefore, this study aimed to investigate the anti-fatigue and antioxidant effects and mechanisms of FA. C2C12 and HepG2 cells were stimulated with 1 mM of hydrogen peroxide (H_2_O_2_) to induce oxidative stress, followed by treatment with FA. Additionally, 44-week-old C57BL/6J mice were orally administered FA for 4 weeks. FA treatment (5–100 μg/mL) significantly attenuated H_2_O_2_-induced cytotoxicity and reactive oxygen species (ROS) production in both cell lines in a dose-dependent manner. In vivo experiments showed that FA treatment significantly increased the mobility time of mice in the forced swimming test and significantly downregulated the serum levels of alanine aminotransferase (ALT), alkaline phosphatase (ALP), lactate dehydrogenase (LDH), creatine kinase (CK), and lactate. Notably, FA treatment significantly upregulated the activities of the antioxidant enzymes superoxide dismutase (SOD), catalase (CAT), and glutathione/oxidized glutathione ratio (GSH/GSSG) and increased the mRNA expression of antioxidant genes (*SOD1*, *SOD2*, *CAT*, *GPx1*, *GPx2*, and *GSR*) in the liver. Conclusively, FA is a potentially useful functional food ingredient for improving fatigue through its antioxidant effects.

## 1. Introduction

Fatigue can be classified as mental fatigue, which leads to a decline in cognitive and perceptual abilities, or physical fatigue, which is characterized by a decrease in muscle capacity due to repeated exercise over a long period of time [1]. Mental fatigue occurs in the nervous system and affects mental activities, such as attention and motivation. Mental fatigue often occurs when performing long-term tasks that are cognitively demanding, and it can cause symptoms such as cardiovascular disease and hypothyroidism [2]. Physical fatigue, also known as muscle fatigue, is caused by repeated physical activities. Physical fatigue occurs gradually and is characterized by the inability of the muscles to maintain the needed strength during physical activity [3,4]. Research findings indicate that the two types of fatigue are interconnected. For example, physical fatigue exacerbates mental fatigue [5,6]. For this reason, it is important to prevent physical fatigue primarily in order to improve fatigue symptoms. The increase in free radicals is the main cause of physical fatigue. When they increase, lipid peroxidation, glutathione oxidation, and oxidative protein damage occur, resulting in muscle fatigue. Therefore, preventing physical fatigue is closely related to the regulation of free radicals [7].

Free radicals are products of normal cell metabolism that are short-lived and highly reactive, causing cell damage through chain reactions [8]. The common byproducts of cellular redox processes are reactive oxygen species (ROS). Moderate levels of ROS promote tissue repair by activating cell survival, apoptosis, and cell differentiation signaling pathways; however, excessive levels of ROS may delay tissue repair and contribute to the development of cancer, arthritis, aging, and chronic and degenerative diseases [9,10]. Conversely, mammals have antioxidant enzyme systems, including superoxide dismutase (SOD), catalase (CAT), glutathione peroxidase (GPx), and glutathione reductases (GSH and GSSG), and they are capable of scavenging ROS [11,12]. Oxidative stress can be neutralized by endogenous antioxidants produced in the body; however, treatment with exogenous antioxidants may facilitate the process [13]. Endogenous antioxidant systems are incomplete in the absence of exogenous reducing compounds, which play an essential role in this mechanism. Therefore, a steady supply of exogenous antioxidants is important for preventing and ameliorating oxidative stress via the antioxidant system [14].

Notably, the liver plays an important role in maintaining metabolic homeostasis, and an imbalance between antioxidants and oxidants in the liver causes oxidative stress. Increased oxidative stress in the liver affects normal metabolic activity and disrupts endogenous molecular metabolism [15,16]. Moreover, oxidative stress can also cause liver diseases, such as chronic viral hepatitis, alcoholic liver disease, and non-alcoholic steatohepatitis, by regulating pathways that control normal biological functions, such as gene transcription and protein expression. Overall, these liver diseases can cause systemic oxidative stress and damage other organs [17]. In addition, one of the main causes of increasing free radicals in the body is excessive exercise. Increased levels of free radicals are caused by complex metabolic changes in muscles, sarcolemmal function, and calcium dysregulation, resulting in decreased motor ability and physical fatigue [18,19]. Additionally, excessive exercise promotes the production of inflammatory cytokines, such as interleukin-6 (IL-6), tumor necrosis factor α (TNF-α), and interleukin 1 beta (IL-1β), in skeletal muscle [20,21], which can cause poor moods and a loss of appetite and exacerbate physical and mental fatigue [22,23]. As a solution, through increased antioxidant enzyme activity in the liver and promoting muscle regeneration, it can be recovered from oxidative stress-induced damage and withstand stimulation caused by free radicals [24].

Deer antler is widely used and has traditional medicinal value in Asia [25], and its various functions have been studied for a long time. Typical efficacy includes the improvement of muscle strength [26] and cardiac functions [27], as well as anti-tumor [28] properties. Recently, research on the functionality of fermented deer antlers, which increases functional ingredients by fermentation, is steadily increasing. In our previous studies, supplementation with *L. curvatus* HY7602-fermented antlers (FAs) has been shown to suppress a loss of muscle mass and function by modulating genes involved in muscle protein synthesis and degradation, mitochondrial biosynthesis, and muscle regeneration [29,30,31,32]. Importantly, these therapeutic effects are associated with an increase in sialic acid content after fermentation [31]. However, the effects of FA on fatigue and antioxidant capacity in muscles have not been studied. Therefore, this study aimed to investigate the effect and molecular mechanism of FA on exercise-induced physical fatigue, especially the role of the antioxidant systems in the liver. In this study, an in vitro experiment was performed using C2C12, which is used in general studies of muscles for evaluation and is commonly used in oxidative stress-induced muscle research [33]. Also, we used HepG2 cells, which possess and mimic specific functions possessed by human hepatocytes and are widely used in in vitro antioxidant enzyme studies [34]. In an in vivo experiment, aged mice with reduced anti-fatigue and antioxidant activity due to aging were used, referring to previous studies [35,36].

## 2. Results

### 2.1. L. curvatus HY7602-Fermented Antler Inhibits Oxidative Stress Damage in Cells

To investigate the effect of FAs on oxidative stress-induced damage, C2C12 and HepG2 cells were treated with various concentrations of FA for 24 h, followed by treatment with 1 mM of H_2_O_2_ for 1 h to induce oxidative stress. In C2C12 myoblasts, FA treatment significantly inhibited H_2_O_2_-induced cytotoxicity and ROS production at concentrations of 20–100 and 50–100 μg/mL, respectively (Figure 1A,B). In the HepG2 cell, FA treatment significantly attenuated H_2_O_2_-induced cytotoxicity and ROS production at concentrations of 1–100 and 5–100 μg/mL, respectively (Figure 1C,D).

### 2.2. L. curvatus HY7602 Fermented Antler Ameliorates Physical Fatigue

To evaluate the effect of FA on physical fatigue in a mouse model, 44-week-old mice were orally administered FA (120 mg/kg body weight) for 4 weeks. On the last day of the experiment, a forced swimming test was conducted to induce physical fatigue, and the serum levels of physical fatigue-related factors were examined. There was no difference in body weight between the experimental groups during the 4 weeks (Figure 2A). Mobility time was significantly longer (*p* < 0.001) in the *L. curvatus* HY7602 fermented antler group (FA; 7.05 ± 0.83 min) than in the control group (CON; 3.45 ± 1.86 min; Figure 2B). In contrast, there was no significant difference in mobility time between the non-fermented antler (NFA; 5.39 ± 1.93 min) and CON groups. 

Additionally, we examined the serum levels of the liver function-related enzymes alanine aminotransferase (ALT), alkaline phosphatase (ALP), and aminotransferase (AST). Serum ALT and ALP levels were significantly lower (*p* < 0.05) in the FA group (16.13 ± 3.13 and 159.40 ± 13.69 U/L, respectively) than in the CON group (21.22 ± 2.28 and 221.60 ± 18.76 U/L, respectively; Figure 2C,D). However, there was no significant difference in AST levels between the groups (Figure 2E). Furthermore, we investigated the serum levels of the physical fatigue-related enzymes lactate dehydrogenase (LDH), creatine kinase (CK), and lactate. Serum LDH, CK, and lactate levels were significantly lower (*p* < 0.05) in the FA group (141.20 ± 23.63 U/L, 38.20 ± 10.76 U/L, 102.64 ± 17.48 mg/dL, respectively) than in the CON group (199.80 ± 50.63 U/L, 59.60 ± 13.83 U/L, 161.64 ± 32.35 mg/dL, respectively; Figure 2F–H). Meanwhile, serum ALP and lactate levels were significantly lower in the NFA group (180.80 ± 13.99 and 119.24 ± 10.57) than in the CON group, and serum ALP and LDH levels were significantly lower (*p* < 0.05) in the FA group than in the NFA group (Figure 2D,F). 

### 2.3. L. curvatus HY7602 Fermented Antler Upregulates Antioxidant-Related Genes and Antioxidant Enzyme Activity in the Liver

To determine the effect of FA on the antioxidant system in the liver during exercise, we examined changes in antioxidant enzyme activity and gene expression in the liver tissue of mice. Real-time polymerase chain reaction (RT-PCR) showed that FA treatment significantly upregulated (*p* < 0.05) the expression of the antioxidant-related genes superoxide dismutase 1 (*SOD1*), (B) superoxide dismutase 2 (*SOD2*), (C) catalase (*CAT*), glutathione peroxidase 1 (*GPx1*), glutathione peroxidase 2 (*GPx2*), and glutathione reductase (*GSR*). Also, SOD2 and CAT mRNA expression levels were significantly upregulated in the NFA group compared with that in the CON group (Figure 3A–F).

Consistent with the results of the RT-PCR, the oral administration of FA significantly increased (*p* < 0.05) the activities of SOD, CAT, and GSH/GSSG by 11.82, 25.45, and 34.80%, respectively, compared with that in the CON group (Figure 4A–C), while unlike the results of RT-PCR, the activities of SOD, CAT, and GSH were not significantly affected in the NFA group (Figure 4A–C).

### 2.4. L. curvatus HY7602 Fermented Antler Promotes Muscle Regeneration

To confirm the effect of FA on muscles during exercise-induced fatigue, the mRNA expression of genes related to muscle fiber regeneration and muscle mass was examined in gastrocnemius and soleus. The calf muscles of mice were separated immediately after sacrifice and weighed. Mice in the FA group (1.13 ± 0.11%) had significantly heavier (*p* < 0.005) calf muscle weight than those in the CON group (1.08 ± 0.08%; Figure 5A). Additionally, the oral administration of FAs significantly upregulated (*p* < 0.05) the expression of the muscle fiber regeneration-related genes myogenic factor 5 (*Myf5*), myogenic differentiation protein 1 (*MyoD*), myosin heavy chain 1 (*Myh1*), and myogenin (*Myog*) (Figure 5B). On the other hand, both calf muscle weight and muscle fiber regeneration-related gene expression were not significantly affected in the NFA group (Figure 5A,B).

### 2.5. L. curvatus HY7602 Fermented Antler Downregulates the mRNA Expression of Cytokines in Muscle

Excessive exercise increases the levels of inflammatory cytokines in muscles, which is strongly related to fatigue [20]. Therefore, we examined the mRNA expression of cytokines in the muscle to confirm the effect of FAs on inflammatory response in muscles during exercise-induced fatigue. The oral administration of FA significantly downregulated (*p* < 0.05) the mRNA expression of cytokines interleukin 1 beta (IL-1β), interleukin 6 (IL-6), and tumor necrosis factor α (TNF-α) (Figure 6A–C). In contrast, the mRNA expression of the cytokines was not significantly affected in the NFA group (Figure 6A–C).

## 3. Discussion

Fatigue has become a chronic symptom in modern society due to lack of exercise, increased work, and mental stress. Fatigue, a nonspecific and subjective symptom, can greatly interfere with daily life and cause diseases; however, accurate diagnosis is difficult. Notably, the symptoms of fatigue are commonly found in patients with various diseases [37,38]. Recently, the role of functional foods in managing these symptoms has attracted the attention of research [39,40]. Our previous study demonstrated that FA treatment effectively improved the loss of muscle function and mass (sarcopenia) [32]. In this study, we investigated the efficacy of FA in the treatment of physical fatigue. Specifically, we examined the antioxidative and muscle regenerative effects of FA using in vitro (C2C12 and HepG2 cells) and in vivo (mice) experiments. 

ROS production, which is induced by both endogenous and exogenous stimuli, plays an important role in cellular physiological processes. Excessive ROS levels delay tissue repair by affecting cell survival and apoptosis-related signaling pathways [41]. Notably, intense exercise-induced increases in ROS production in the muscles can cause oxidative stress, fatigue, and cell damage and increase physical fatigue [42,43]. In the present study, the FA treatment (5–100 μg/mL) significantly inhibited H_2_O_2_-induced cytotoxicity and ROS production in C2C212 myoblasts and HepG2 cells (Figure 1A–D). ROS are also produced in the liver, and an imbalance caused by a decrease in intracellular antioxidant defense systems or the overproduction of ROS can lead to oxidative stress [44]. Importantly, oxidative stress contributes to the pathogenesis of several liver diseases, such as fibrosis and cancer, and patients with these liver diseases often experience severe fatigue [45]. Collectively, these results indicate that FA ameliorates oxidative stress-induced damage and fatigue at the cellular level. 

To confirm the results of the in vitro study, in vivo experiments were conducted using mice. The oral administration of FAs increased the mobility time of mice in the forced swimming test compared to the control group (Figure 2B). The forced swimming test is conducted to induce physical fatigue, and it is a representative experiment for evaluating exercise-induced fatigue and endurance. Exercise-induced physical fatigue after a forced swimming test was confirmed by examining the serum levels of ALT, AST, ALP, LDH, CK, and lactate. The serum levels of ALT, AST, and ALP are indicators of liver damage, and they increase during liver cell damage or inflammation. Importantly, vigorous exercise can also increase the serum levels of these enzymes [46,47]. LDH and CK levels are closely related to muscle damage. Previous studies have reported elevated blood levels of LDH and CK after high-intensity aerobic exercise [48]. Lactate is an indicator of muscle fatigue, which can increase because of the changes in glycolysis caused by repeated muscle contractions [49]. In the present study, the FA treatment significantly decreased the levels of ALT, ALP, LDH, CK, and lactate (Figure 2C–H), suggesting that the treatment may improve exercise endurance [50]. Moreover, the increased mobility time in the FA group was associated with increased serum levels of ALP, ALT, LDH, CK, and lactate. Similarly, the oral administration of NFA significantly decreased the serum levels of ALP and lactate but did not affect the other parameters. Meanwhile, the AST level decreased slightly in the FA group, but there was no significant difference. Previous studies have shown that not all serum levels of indicators have the same tendency [51]. However, further studies are needed as to why exactly FFA does not affect AST levels.

Enzymatic and non-enzymatic antioxidant systems are essential for cellular responses to oxidative stress. Antioxidant enzymes, such as SOD, CAT, and GPx, as well as non-enzymatic electron receptors, such as GSH, can contribute to the antioxidant system of the liver, thereby reducing increased oxidative stress [52]. The oral administration of FA significantly increased the mRNA expression of antioxidant enzyme-related genes in the liver (Figure 3A–F). Among these, SOD and GSH are functions that protect cells from the toxic effects of endogenously generated free radicals [53,54], and endogenous free radicals produced by oxidative stress promote cell damage and inflammatory reactions [23,55]. Therefore, increased *SOD* and *GSH* levels in the FA group are thought to be associated with decreased indicators related to tissue damage or inflammation in the serum. Moreover, the H_2_O_2_ scavenging enzymes GPx2 and CAT [56] showed the highest increase, which was consistent with the results of the in vitro cytotoxicity and ROS production assays. Similarly, SOD2 and CAT mRNA expression increased significantly in the NFA group. Furthermore, we examined the activities of antioxidant enzymes, such as SOD, CAT, and GSH/GSSG, in the liver. These enzymes are involved in ROS scavenging [56]. The oral administration of the FA extract significantly increased SOD, CAT, and GSH/GSSG enzymatic activities in the liver (Figure 4A,B). Overall, the increase in the levels of antioxidant enzymes in the liver may contribute to the detoxification of ROS produced during exercise, resulting in a decrease in the mRNA expression of oxidative stress-related factors. Additionally, the NFA group only showed significant differences in the expression of a few antioxidant-related genes, which was consistent with the results of mobility time.

Excessive exercise not only causes muscle fatigue but also leads to a loss of muscle mass. At this time, the greater the number of myofibroblasts, the better the ability to reconstruct skeletal muscle and the faster the ability to recover after damage [57]. In a previous study, we confirmed that FA promoted muscle fiber regeneration in mice to improve muscle mass [31,32]. Similarly, the FA treatment increased the muscle mass of mice and upregulated the expression of genes related to muscle fiber regeneration (*Myf5*, *MyoD*, *Myh1*, and *Myog*; Figure 5A,B). Furthermore, the levels of cytokines in the muscles are also associated with fatigue. Proinflammatory cytokines such as TNF-α, IL-1, IL-6, and IL-12 cause inflammation, which exacerbates fatigue [58]. IL-1β upregulation has been shown to inhibit muscle fiber differentiation and regeneration [59]. The FA treatment downregulated IL-1β mRNA expression, which was consistent with the upregulation of muscle fiber regeneration-related genes in the FA group (Figure 6A). Additionally, the FA treatment downregulated the mRNA expression of IL-6 and TNF-α in the muscle (Figure 6B,C). Excessive muscle contraction induces the secretion of IL-6, and TNF-α interferes with the muscle contraction function, which may affect exercise performance [58,60]. Collectively, these results suggest that the decrease in the expression of proinflammatory cytokines may increase the mobility time of mice in the swimming test. However, the correlation between fatigue-related changes in the muscle and liver is still unknown.

Conclusively, In vivo experiments showed that FA oral administration upregulates both the mRNA and protein levels of antioxidant enzymes, which may ameliorate oxidative stress and improve exercise performance. Moreover, FA suppresses the serum levels of factors related to exercise-induced physical fatigue, upregulates the expression of genes related to muscle fiber regeneration, and suppresses the mRNA expression of proinflammatory cytokines. However, the correlation between the liver and muscle and its effects on mental fatigue remains unknown. Therefore, it is unclear whether FA can affect mental fatigue. Further studies should examine the effects of fermented antler-induced improvements on physical fatigue on mental fatigue.

## 4. Materials and Methods

### 4.1. Preparation of L. curvatus HY7602-Fermented Deer Antler

The deer antler used in this study was derived from Cervus elaphus Linné, purchased from Yoojintongsang (Seoul, Republic of Korea), and it was used in accordance with the standards for herbal medicine (Korean Pharmacopoeia, Ministry of Food and Drug Safety). The middle and lower parts of the dried deer antler were thinly sliced and extracted with distilled water (1:30, *w*/*w*) at 95 ± 5 °C until Brix reached 1.0 ± 0.5. For fermentation, *L. curvatus* strain HY7602 freeze-dried powder (hy Pyeongtaek Factory Co., Ltd., Pyeongtaek-si, Republic of Korea) (1%, *w*/*w*) was mixed with the antler extract and incubated at 37 ± 2 °C for 24 ± 1 h. After incubation, the mixture was pasteurization at 95 ± 5 °C for 20 ± 2 min and concentrated to 21 ± 1.0° Brix (Batch Type Evaporator, JiangSu Shijie Technology Co., Ltd., Nanjing, China). Finally, the mixture was mixed with maltodextrin and lyophilized to obtain the fermented antler powder. 

### 4.2. In Vitro Studies

#### 4.2.1. Cell Culture

C2C12, a mouse skeletal muscle myoblast, and HepG2, a human hepatic cell line, were obtained from the Korean Cell Line Bank (KCLB, Seoul, Republic of Korea) and cultured in high-glucose Dulbecco’s modified Eagle medium (DMEM) containing 10% fetal bovine serum and 1% penicillin streptomycin (Gibco, Grand Island, NY, USA).

#### 4.2.2. Measurement of Cell Viability

The viability of hydrogen peroxide (H_2_O_2_)-treated C2C12 myoblast and HepG2 cells was assessed using the CytoTox 96^®^ Non-Radioactive Cytotoxicity Assay Kit (Promega, Madison, WI, USA), according to the manufacturer’s instructions with slight modifications. The experimental method was as follows. Cells were incubated in 96-well plates containing the growth medium until confluency reached 80%. Thereafter, cells were cultured in serum-free DMEM with or without (control) FA (1, 5, 10, 20, 50, and 100 μg/mL) for 24 h, followed by incubation with 1 mM of H_2_O_2_ (Sigma-Aldrich, St. Louis, MO, USA) for an additional 1 h. After incubation, 40 μL of supernatant from each well was mixed with 40 μL of Cytotox 96 reagent and allowed to stand for 30 min. Absorbance was measured at 490 nm using a BioTek Synergy HTX multimode reader (Agilent, Santa Clara, CA, USA). Data are expressed as the mean ± standard deviation (SD) of three independent experiments.

#### 4.2.3. Determination of ROS Levels

ROS levels in C2C12 myoblasts and HepG2 cells were determined according to a previous method [61] with slight modifications. The experimental method was as follows. Cells were incubated in 96-well plates containing the growth medium until confluency reached 80%. Thereafter, cells were cultured in serum-free DMEM with or without (control) FA (1, 5, 10, 20, 50, and 100 μg/mL) for 24 h, followed by incubation with 200 μM of H_2_O_2_ for 1 h. After removing the medium, 100 μL of 10 μM H_2_DCFDA was added to each well, followed by incubation for 20 min. All reagents were discarded, and fluorescence was measured at an excitation of 480 nm and emission at 530 nm using a BioTek Synergy HTX multi-mode reader (Agilent, Santa Clara, CA, USA). Data are expressed as the mean ± standard deviation (SD) of three independent experiments.

### 4.3. Animals and Diets

Male ICR mice (44-week-old; *n* = 18) were purchased from DooYeol Biotech (Seoul, Republic of Korea). The mice were housed individually in a controlled environment (temperature of 20–22 °C, humidity of 40–60%, and 12 h light/dark cycle) and fed a rodent diet (crude protein: 18.4%; fat: 6.0%; carbohydrates: 44.2%; crude fiber: 3.8%; neutral detergent fiber: 14.7%; ash: 5.5%; Envigo, Indianapolis, IN, USA) for 1 week during the acclimation period. Thereafter, the mice were divided into three groups (*n* = 6): old mice (CON), old mice administered NFA, and old mice administered FA. Mice in the CON, NFA, and FA groups were orally administered saline, NFA, and FA, respectively, once a day for 4 weeks (120 mg/kg body weight). Bodyweight and feed intake were measured weekly. The animal study was conducted according to the guidelines of hy Co., Ltd. (Yongin-si, Republic of Korea) and approved by the Institutional Animal Care and Use Committee of hy Co., Ltd. (IACUC approval number, AEC-2023-0002-Y).

### 4.4. Forced Swimming Test

The forced swimming test was performed immediately before sacrifice. The test was conducted at 25 ± 1 °C, with water to a depth of 22 cm in a transparent cylindrical acrylic tank (25 cm height × 14 cm diameter) until exhaustion. Exhaustion was defined as the inability of the mice to rise to the surface to breathe within 5 s. Total mobility time was expressed as the average of the group. 

### 4.5. Sample Collection and Serum Biochemistry

At the end of the study, liver, calf muscle, and whole blood samples were collected for various analyses. The liver tissue was stored at −80 °C immediately upon collection. The calf muscle tissue was weighed, divided into gastrocnemius (GA) and soleus (SOL), and stored at −80 °C. Whole blood samples were collected via cardiac puncture, followed by centrifugation at 2000× *g* for 15 min at 4 °C to separate the serum, which was stored at −80 °C until further use. Serum levels of ALT, AST, ALP, LDH, CK, and lactate were measured using an automated analyzer (Hitachi 7020; Hitachi, Ltd., Tokyo, Japan).

### 4.6. Real-Time Polymerase Chain Reaction (RT-PCR)

The total RNA was extracted from liver tissue (100 mg) and SOL (20 mg) using an easy-spin Total RNA Extraction Kit (iNtRON Biotechnology, Gyeonggi, Republic of Korea), followed by cDNA synthesis from RNA (2 μg) using the Omniscript RT Kit (QIAGEN, Hilden, Deutschland). The cDNA template was used for RT-PCR. Gene expression analysis was performed using the QuantStudio 6 RT-PCR program (Thermo Fisher Scientific, Waltham, MA, USA). The quantifications of superoxide dismutase 1 (SOD1, Mm01344233_g1), superoxide dismutase 2 (SOD2, Mm01313000_m1), glutathione peroxidase 1 (GPx1, Mm00656767_g1), glutathione peroxidase 2 (GPx2, Mm00850074_g1), citrate synthase (CS, Mm00466043_m1), catalase (CAT, Mm00437992_m1), glutathione reductase (GSR, Mm00439154_m1), myogenic factor 5 (Myf5, Mm00435125_m1), myoblast determination protein 1 (MyoD, Mm_00440387_m1), type 2×myosin heavy chain (Myh1, Mm01332489_m1), myogenin (Myog, Mm00446194_m1), interleukin 1 beta (Il1b, Mm00434228_m1), interleukin 6 (Il6, Mm00446190_m1), tumor necrosis factor (Tnf, Mm00443258_m1), and glyceraldehyde-3-phosphate dehydrogenase (GAPDH, Mm_99999915_g1) were performed using gene-specific primers purchased from Applied Biosystems (Middlesex Country, MA, USA). The expression levels of the target genes were normalized to that of GAPDH using the comparative *C*_T_ method.

### 4.7. Evaluation of Antioxidant Enzyme Activities

The activities of SOD1, CAT, and GSH/GSSG were measured using an oxidative stress bioassay kit (BIOMAX, Gyeonggi, Republic of Korea) according to the manufacturer’s instructions. For the SOD1 activity assay, liver tissue (100 mg) was washed with PBS and homogenized in sucrose buffer, followed by centrifugation (4 °C, 10,000× *g*, 60 min) to obtain the supernatant for analysis. For the CAT activity assay, the liver tissue (50 mg) was homogenized with a 1×Assay Buffer, debris was removed, and the sample was used for analysis. For the GSH/GSSG activity assay, liver tissue (100 mg) was homogenized in ice-cold 5% MPA, and the supernatant was used for analysis.

### 4.8. Statistical Analysis

All data are expressed as mean ± SD. The significance difference between the control and treatment groups was determined using an unpaired Student’s *t*-test. In the in vitro study, statistical differences were compared with the H_2_O_2_-only group. All statistical analyses were performed using SPSS version 26.0 (IBM Corp., Armonk, NY, USA).

## Figures and Tables

**Figure 1 ijms-25-03318-f001:**
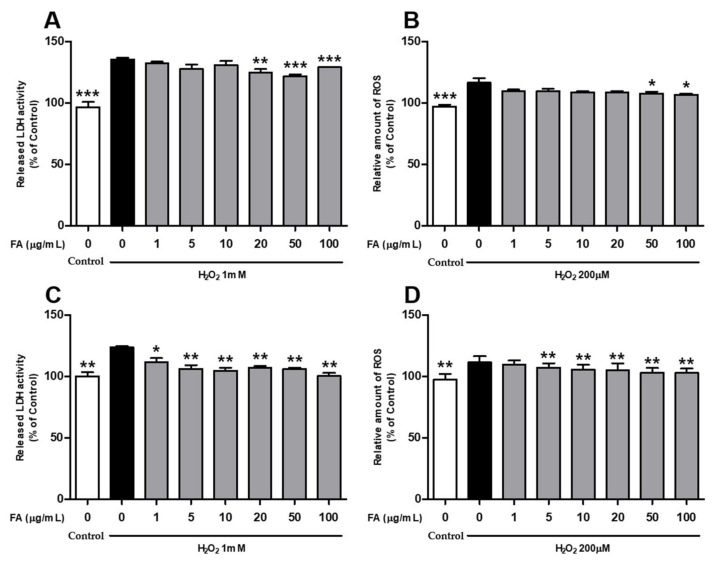
Effect of *L. curvatus* HY7602 fermented antlers (FAs) on oxidative stress-induced damages in C2C12 myoblasts and HepG2 cells. Cells were treated with 0, 1, 5, 10, 20, 50, and 100 μg/mL of FA for 24 h. (**A**) Lactate dehydrogenase (LDH) and (**B**) reactive oxygen species (ROS) levels in C2C12 myoblasts after 1 h of incubation with 1 mM of H_2_O_2_. (**C**) LDH and (**D**) ROS levels in HepG2 cells after 1 h of incubation with 1 mM H_2_O_2_. Data are expressed as the mean ± standard deviation (SD) of three independent experiments. Statistical analysis was performed using unpaired Student’s *t*-tests. Statistical differences compared with the H_2_O_2_-only group at *p* < 0.05, <0.01, and <0.001 are indicated by *, **, and ***, respectively.

**Figure 2 ijms-25-03318-f002:**
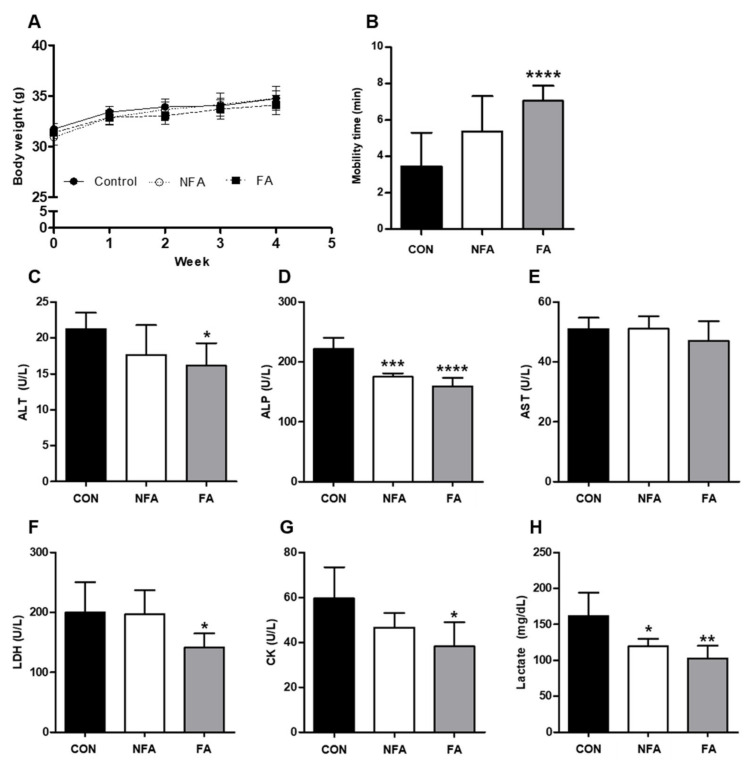
Effect of *L. curvatus* HY7602 fermented antlers (FAs) on exercise-induced physical fatigue. (**A**) Body weight was measured weekly. (**B**) Mobility time during the forced swimming test was performed on the last day of the experiment. Serum (**C**) alanine aminotransferase (ALT), (**D**) alkaline phosphatase (ALP), (**E**) aminotransferase (AST), (**F**) lactate dehydrogenase (LDH), (**G**) creatine kinase (CK), and (**H**) lactate levels were measured. Data are expressed as the mean ± standard deviation (SD; *n* = 6 mice/group). Statistical analysis was performed using unpaired Student’s *t*-tests. Statistical differences compared with the CON group at *p* < 0.05, <0.01, <0.005, and <0.001 are indicated by *, **, ***, and ****, respectively. CON, old mice; NFA, old mice treated with non-fermented antler; FA, old mice treated with *L. curvatus* HY7602-fermented antler.

**Figure 3 ijms-25-03318-f003:**
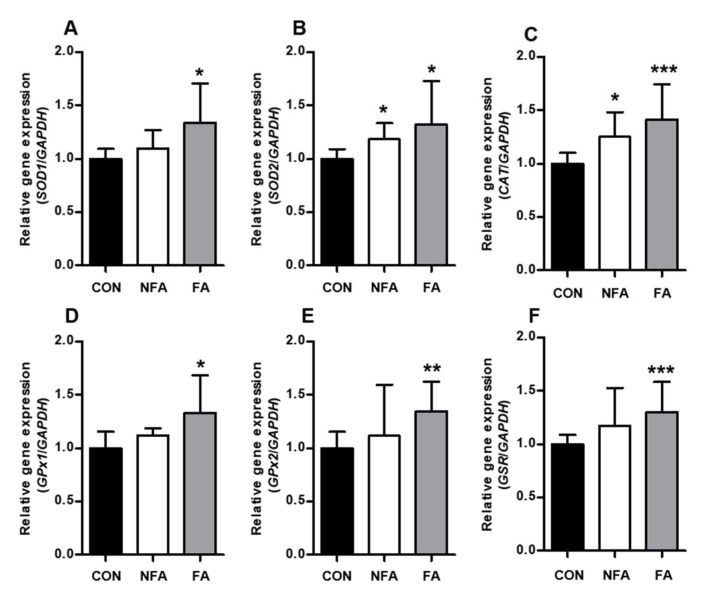
Effect of *L. curvatus* HY7602 fermented antlers (FAs) on the expression of antioxidant-related genes in the liver. Real-time polymerase chain reaction (RT-PCR) was performed to determine the mRNA expression of (**A**) superoxide dismutase 1 (*SOD1*), (**B**) superoxide dismutase 2 (*SOD2*), (**C**) catalase (*CAT*), (**D**) glutathione peroxidase 1 (*GPx1*), (**E**) glutathione peroxidase 2 (*GPx2*), and (**F**) glutathione reductase (*GSR*). Data are expressed as the mean ± standard deviation (SD; *n* = 6 mice/group). Statistical analysis was performed using unpaired Student’s *t*-tests. Significant differences compared with the CON group at *p* < 0.05, <0.01, and <0.005 are indicated by *, **, and ***, respectively. CON, old mice; NFA, old mice treated with non-fermented antler; FA, old mice treated with FA.

**Figure 4 ijms-25-03318-f004:**
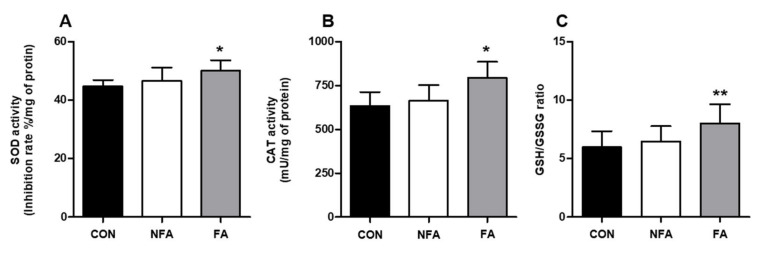
Effect of *L. curvatus* HY7602 fermented antler (FAs) on antioxidant-enzyme activity in the liver. The activity of (**A**) superoxide dismutase (SOD), (**B**) catalase (CAT), and (**C**) glutathione (GSH/GSSG ratio) in the liver tissue was measured. Data are expressed as the mean ± standard deviation (SD; *n* = 6 mice/group). Statistical analysis was performed using unpaired Student’s *t*-tests. Significant differences compared with the CON group at *p* < 0.05 and <0.005 are indicated by * and **, respectively. CON, old mice; NFA, old mice treated with non-fermented antler; FA, old mice treated with FA.

**Figure 5 ijms-25-03318-f005:**
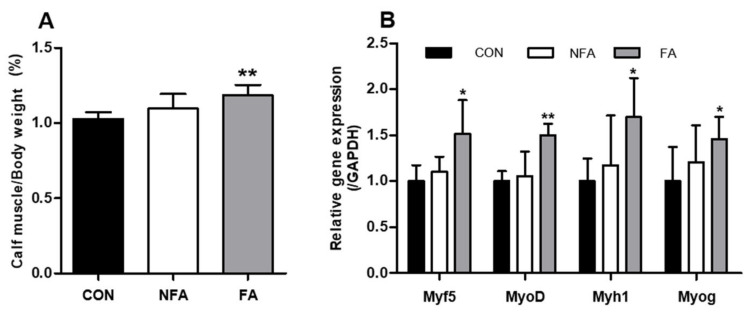
Effect of *L. curvatus* HY7602 fermented antler (FA) on muscle regeneration. (**A**) Muscle weight (gastrocnemius and soleus) was measured. (**B**) The expression of the muscle fiber regeneration-related genes myogenic factor 5 (*Myf5*), myoblast determination protein 1 (*MyoD*), myosin heavy chain 1 (*Myh1*), and myogenin (*Myog*) was detected using real-time polymerase chain reaction (RT-PCR). Data are expressed as the mean ± standard deviation (SD; *n* = 6 mice/group). Statistical analysis was performed using unpaired Student’s *t*-tests. Significant differences compared with the CON group at *p* < 0.05 and < 0.005 are indicated by * and **, respectively. CON, old mice; NFA, old mice treated with non-fermented antler; FA, old mice treated with FA.

**Figure 6 ijms-25-03318-f006:**
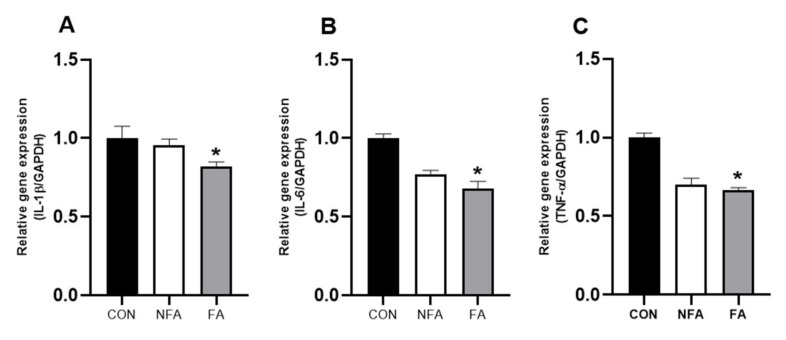
Effect of *L. curvatus* HY7602 fermented antler (FA) on the mRNA expression of cytokines in muscles. The expression of (**A**) interleukin 1 beta (*IL-1β*), (**B**) interleukin 6 (*IL-6*), and (**C**) tumor necrosis factor α (*TNF-α*) was detected using real-time polymerase chain reaction (RT-PCR). Data are expressed as the mean ± standard deviation (SD; *n* = 6 mice/group). Statistical analysis was performed using unpaired Student’s *t*-tests. Significant differences compared with the CON group are indicated by * at *p* < 0.05. CON, old mice; NFA, old mice treated with non-fermented antler; FA, old mice treated with FA.

## Data Availability

Data are contained within the article.

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
