# Peer review of "Effect of HY7602 Fermented Deer Antler on Physical Fatigue and Antioxidant Activity in Mice"

_ijms, 2024, doi:10.3390/ijms25063318_

Round 1
Reviewer 1 Report
Comments and Suggestions for Authors
Please see the attachement.

Reviewer 2 Report
Comments and Suggestions for Authors
The manuscript is interesting and well-written, with a clear research rationale. The following comments are to be considered:
Concentrations of H2O2 and fermented antler (best ones) are better listed in the abstract. Also, make sure they appear clearly in the subsequent sections of the paper.
Line 83,84: refer briefly to the identity of each of the two used cell lines. Also add the used H2O2 to this paragraph.
Line 100: add FA concentration.
Line 103: fix the word case of “mobility”.
The position of figures 2D and 2E should be reversed on Figure 2 and lines 112, 113.
Lines 136-139: gene names should be italicized. Consistent gene names should be used throughout the manuscript.
Line 154: change “by the NFA group” to “in the NFA group”.
Line 168: do you mean by calf muscles gastrocnemius and soleus? If so, list in parentheses.
Comments on the Quality of English LanguageThe of the manuscript is acceptable.
Reviewer 3 Report
Comments and Suggestions for Authors
In this study, the anti-fatigue and antioxidant effects of Lactobacillus curvatus HY7602-fermented antler (FA) were investigated. The results showed that FA treatment attenuated oxidative stress-induced cytotoxicity and reactive oxygen species (ROS) production in cell lines. In vivo experiments also demonstrated that FA increased the mobility time of mice and downregulated the levels of enzymes associated with fatigue. Additionally, FA upregulated the activities of antioxidant enzymes and increased the expression of antioxidant genes in the liver. These findings suggest that FA could be a beneficial functional food ingredient for combating fatigue through its antioxidant effects.
Comments:
Introduction: The first two paragraphs are poorly connected. The first paragraph talks about fatigue, while the second paragraph mentions oxidative stress. A smoother transition should be made to explain how these two processes are related. Overall, the introduction does not effectively introduce the reader to the specific problem addressed in the study. It should be rewritten to delve deeper into the specific issues, rather than just providing a general description of fatigue, oxidative stress, and inflammation.
Results: There are doubts about the reliability of the differences between samples in some figures, such as Figure 3, considering the large overlap in confidence intervals between the graphs. The authors could provide raw data as supplementary material to increase confidence in their results.
Reviewer 4 Report
Comments and Suggestions for Authors
1. The introduction provides a broad overview of fatigue, its types, and the role of oxidative stress in fatigue and related diseases. However, it could benefit from a more direct connection to the specific focus of the study on fermented deer antler and its potential effects. Clarifying how existing research gaps directly lead to the current study could strengthen the rationale.
2. While the introduction mentions the interconnectedness of physical and mental fatigue, it does not elaborate on whether the study will address mental fatigue aspects, potentially leaving the scope of the study somewhat unclear.
3. The introduction mentions that physical and mental fatigue are interconnected, then mentions an example of how physical fatigue exacerbates mental fatigue. I think the example should be more specific, such as the role of physical fatigue and ROS accumulation plays a role in neurodegenerative diseases including ALS, as mentioned in this review article https://doi.org/10.1016/j.ejmech.2024.116151
4. It is mentioned in 4.2.2 and 4.2.3 that you followed the manufacturer’s instructions with slight modifications. Please mention the modifications.
5. Clarify the source and quality of the sliced antlers (Cervus elaphus Linné) used for extraction. Additionally, provide more details on the extraction process, such as the duration of extraction and any specific equipment used.
6. Specify the concentration of the L. curvatus strain HY7602 freeze-dried powder added to the antler extract (1%, w/w). Additionally, elaborate on the rationale behind selecting this specific concentration.
7. Describe the rationale behind selecting C2C12 myoblasts and HepG2 cells as model systems for studying anti-fatigue and antioxidant effects.
8. Provide justification for selecting the concentrations of FA (1, 5, 10, 20, 50, and 100 μg/mL) used in the experiments, including any preliminary dose-response studies conducted.
9. Explain the reasoning behind using 1 mM of H2O2 to induce oxidative stress in the cells and how this concentration was determined.
10. Justify the choice of male ICR mice aged 44 weeks for the in vivo experiments, including any relevance to the study's objectives or previous findings.
11. Clarify the rationale behind selecting the dosage of FA (120 mg/kg body weight) for oral administration to the mice, including any considerations for toxicity or previous efficacy studies.
12. Provide more information on the pasteurization process applied to the mixture after fermentation, including the temperature and duration of pasteurization.
13. If applicable, a comparison with control groups or existing treatments in detail could provide a clearer picture of the fermented deer antler's efficacy relative to other interventions. 14. While the article focuses on the beneficial effects of FA, discussing potential side effects or contraindications, if any, could provide a more comprehensive risk-benefit analysis. 15. The manuscript mentioned reactive nitrogen species in the introduction but did not test the effect of FA on them. Does this imply that the effect might be similar to that on reactive oxygen species, or does it indicate a need for further studies? 16. In the results from line 85 to 89, the study demonstrated the effect of FA on H2O2-induced cytotoxicity and ROS production. Interestingly, in HepG2 cells, the concentration required for inhibition was lower than that in C2C12 myoblasts. It would be beneficial to mention some possible reasons for this discrepancy. 17. While you briefly mention the role of antioxidant enzymes in ameliorating oxidative stress, consider expanding on the underlying mechanisms by which FA exerts its effects. Providing more detailed explanations of how FA influences cellular processes related to oxidative stress and fatigue would enhance the scientific rigor of the discussion. 18. The discussion does not mention the AST levels when discussing the results. It was only mentioned that the increase of its serum level is an indicator of liver damage, but there’s no mention of AST levels in the sample. Is there a reason why AST levels didn’t alter significantly when FA was administered? 19. In Figure 2A the lines are not clear. Either use different colors or make the graph larger to make it easier for viewers to differentiate between the datasets. 20. Reference 37 is not referenced in the same format as the others. Please ensure all references are formatted consistently.
Round 2
Reviewer 1 Report
Comments and Suggestions for Authors
The authors answered to all my questions and suggestions. Manuscript can be published.
Reviewer 3 Report
Comments and Suggestions for Authors
Accept in the present form
Reviewer 4 Report
Comments and Suggestions for Authors
Thank you for addressing all of the comments.